# Focal Therapy Using High-Intensity Focused Ultrasound for Low- and Intermediate-Risk Prostate Cancer: Results from a Prospective, Multicenter Feasibility Trial

**DOI:** 10.3390/cancers17213429

**Published:** 2025-10-25

**Authors:** Gabor Rosta, Simon Turba, Dong-Ho Mun, Azad Shehab, Leon Saciri, Paul F. Engelhardt, Patricia Weisz, Claus Riedl, Ghazal Ameli, Stephan Doblhammer, Harun Fajkovic

**Affiliations:** 1Division of Urology and Andrology, University Hospital St. Pölten, 3100 St. Pölten, Austria; simon.turba@stpoelten.lknoe.at (S.T.); azad.shehab@stpoelten.lknoe.at (A.S.); leon.saciri@stpoelten.lknoe.at (L.S.); 2Karl Landsteiner University of Health Sciences, 3500 Krems, Austria; 3Urology Department, Landesklinikum Baden, 2500 Baden, Austria; paul.engelhardt@aon.at (P.F.E.); patricia.weisz@baden.lknoe.at (P.W.); claus.riedl@baden.lknoe.at (C.R.); 4Paracelsus Private University, 5020 Salzburg, Austria; 5Urology Department, Landesklinikum Korneuburg, 2100 Korneuburg, Austria; ghazal.ameli@korneuburg.lknoe.at (G.A.); stephan.doblhammer@korneuburg.lknoe.at (S.D.)

**Keywords:** prostate cancer, focal therapy, high-intensity focused ultrasound

## Abstract

**Simple Summary:**

Whole-gland surgery or radiotherapy for early prostate cancer can cure the disease but often leave men with difficulty passing urine, bladder leakage, or reduced sexual function. Our study tested a gentler method that destroys only the tumor area with a precisely aimed beam of ultrasound. Fifty-one patients with low- or intermediate-risk prostate cancer were treated at three hospitals and followed for two years. Most men required no additional radical treatment, and follow-up biopsies were negative in more than 80% of cases. Side effects were mild and short-lived, and quality-of-life scores remained stable. These results suggest that targeted focused ultrasound could offer an effective, organ-sparing alternative for carefully selected patients and justify larger, long-term studies.

**Abstract:**

**Background/Objectives**: Whole-gland surgery or radiotherapy for localized prostate cancer (PCa) can cure the disease but often impair urinary and sexual function. Focal therapy with high-intensity focused ultrasound (HIFU) seeks to eradicate the tumor while sparing uninvolved tissue. We prospectively evaluated oncological control, functional outcomes and safety of MRI-guided focal HIFU in patients with low- or intermediate-risk PCa. **Methods**: In this prospective, single-arm, phase II feasibility trial (three Austrian centres, 2021–2024), treatment-naive patients with D’Amico low/intermediate-risk, PSA ≤ 15 ng/mL, clinical stage ≤ T2 and MRI-targeted, biopsy-confirmed index lesions underwent lesion-targeted HIFU (Focal One™). The primary endpoint was failure-free survival (FFS: absence of salvage whole-gland or systemic therapy, metastasis or PCa-specific death). Secondary endpoints included biopsy-proven cancer, prostate-specific antigen (PSA), patient-reported symptoms as International Prostate Symptom Score (IPSS), 5-item International Index of Erectile Function (IIEF), Gaudenz Incontinence Questionnaire and adverse events. Planned follow-up was 24 months with PSA every 3 months, mpMRI and biopsies at 12 months, and imaging- or PSA-triggered biopsies thereafter. **Results**: Fifty-one men were analysed in the per-protocol cohort (median age 67 years, median PSA 7.55 ng/mL). Median treated volume was 12 mL; median procedure time 85 min. At 24 months, FFS was 94.1%: 3/51 patients (5.9%) required salvage radiotherapy. Among 31 patients who underwent follow-up biopsy, 26 (83.9%) had no cancer; the five positives included three ISUP 1, one ISUP2 and one ISUP 4 lesion. Mean PSA fell by 69% at 3 months (to 2.3 ng/mL) and then stabilized under 3 ng/mL, with a mean of 2.7 ± 1.5 ng/mL at 24 months. Transient acute urinary retention occurred in 11/51 (21.6%); no Clavien–Dindo grade ≥ 4 events were reported. IPSS returned to or improved beyond baseline, erectile function largely recovered by 6–12 months, and only one new case of grade 2 incontinence was observed. **Conclusions**: MRI-guided focal HIFU achieved high two-year failure-free survival with low morbidity and preserved quality of life in carefully selected patients with low- or intermediate-risk PCa. These data support further randomized and longer-term investigations of focal HIFU as an organ-sparing alternative to whole-gland treatment.

## 1. Introduction

Prostate cancer (PCa) represents a significant global health issue, necessitating effective and patient-centered treatment approaches [1]. Standard whole-gland treatments, including radical prostatectomy and radiotherapy, remain the primary management strategies for clinically significant prostate cancer but are frequently associated with urinary, sexual, and bowel-related adverse effects [2].

Focal therapy (FT) has emerged as a promising treatment option aimed at achieving comparable oncological outcomes to whole-gland therapies while minimizing treatment-related adverse events (AEs) [3,4,5]. This approach has generated considerable interest among urologists and patients, particularly for the management of low- to intermediate-risk PCa [6]. The growing enthusiasm for FT can largely be attributed to advancements in early detection of localized PCa through effective screening programs and the enhanced diagnostic accuracy provided by multiparametric Magnetic Resonance Imaging (mpMRI) in identifying and localizing clinically significant lesions [1,4,7].

FT primarily employs ablative techniques, with high-intensity focused ultrasound (HIFU), cryotherapy, photodynamic therapy, irreversible electroporation, and focal brachytherapy being among the leading modalities [8]. HIFU, specifically, is currently the most extensively utilized technology, with the largest body of clinical data available [2,9]. Despite its increasing adoption, current European Association of Urology (EAU) prostate cancer guidelines categorize FT as an experimental modality, recommending its application exclusively within clinical trial settings due to a significant lack of robust prospective evidence [10,11,12]. Nonetheless, surveys indicate widespread usage among urologists, with approximately half of European urologists recommending or performing FT [13].

This prospective, single-arm, multicenter feasibility trial was designed to evaluate the practicality and safety of focal HIFU in men with low- to intermediate-risk prostate cancer, with primary emphasis on functional outcomes and adverse events; exploratory oncologic signals were collected but the study was not powered for definitive oncologic endpoints.

## 2. Materials and Methods

This prospective, non-randomized, phase II, feasibility clinical study was conducted at three hospitals in Lower Austria from April 2021 to December 2024. Approval was obtained from the Lower Austria Ethical Board (approval number GS4-EK-3/162-2020). All participants provided written informed consent after receiving comprehensive information regarding established treatment options such as radical prostatectomy (RP) or radiotherapy (RT) and explicitly chose not to pursue active surveillance.

Eligible patients were treatment-naive males diagnosed with non-metastatic PCa categorized as low- or intermediate-risk according to the D’Amico classification [14], with protocol-limited PSA levels ≤ 15 ng/mL and clinical stage ≤ cT2b. Additional inclusion criteria mandated the absence of significant lesions (classified as Prostate Imaging Reporting and Data System Version 2.1 (PI-RADS ≥ 4) [15] on the contralateral prostate side and biopsy-confirmed prostate cancer lesions concordant with mpMRI findings. Low-risk patients were included only if they had declined active surveillance.

Exclusion criteria encompassed prior prostate surgery within the preceding six months, tumor proximity of less than 7 mm to the urethral sphincter, and treatment areas more than 45 mm away from the rectum [16]. Patients presenting with perilesional prostate calcification or rectal pathologies were also excluded. Additionally, patients showing evidence of extraprostatic tumor extension, suspicious regional lymph nodes, or distant metastases identified by mpMRI, cross-sectional imaging, or bone scans were excluded from participation.

### 2.1. Pre-Treatment

Patient characteristics were systematically recorded. These included age at intervention, PSA level, prostate volume, PI-RADS classification, Gleason grade group, International Prostate Symptom Score Questionnaire (IPSS), International Index of Erectile Function Questionnaire (IIEF), Gaudenz Incontinence Questionnaire [17], family history of prostate cancer, current medications, comorbidities, findings from digital rectal examination, and smoking history. Health-related quality of life was assessed using the Medical Outcomes Study Short Form-36 (SF-36) [18].

### 2.2. Treatment Protocol

HIFU procedures were performed using the Focal One system (EDAP TMS, Lyon, France). Ablation was directed at biopsy-confirmed index lesions with ISUP ≥ 2. In patients with low-risk PCa, the ISUP 1 lesions were treated. Treatment planning was based on mpMRI images, with urologists and radiologists collaboratively contouring the prostate. Treatment was monitored in real time via live sonography. A safety margin of 9 mm was applied around all targeted lesions. The neurovascular bundles and urethra were not specifically spared.

Contrast-enhanced ultrasound (CEUS) (SonoVue^®^) was used intraoperatively to assess ablation completeness. If residual tissue was suspected, an additional treatment pass was administered.

A Foley catheter was placed at the end of the procedure. All treatments were performed under general anesthesia.

### 2.3. Adverse Events and Salvage Therapy

Complications were recorded prospectively and graded according to the Clavien–Dindo classification [19]. The protocol permitted repeat HIFU sessions. Patients could undergo radical treatment based on follow-up biopsy findings [20].

### 2.4. Follow-Up

Follow-up included PSA testing every three months during the first two years post-treatment. Patients completed the IPSS, IIEF, and Incontinence Questionnaire at 3, 6, 12, and 24 months. Health-related quality of life was assessed using the SF-36 [18].

Multiparametric MRI was repeated at 12 and 24 months. Follow-up mpMRI/US Fusion biopsies were conducted at 12 months. These included systematic biopsies from both treated and untreated regions, and targeted biopsies if suspicious lesions were visible.

Beyond 12 months, biopsies were performed only when biochemical or imaging findings suggested recurrence, rather than on a fixed timetable [21]. Triggers included PSA rise or suspicious lesions on mpMRI.

Our primary outcome was Failure-Free Survival (FFS) defined as no transition to any of the following: local salvage therapy (radical prostatectomy or radiotherapy), systemic therapy, metastases, or prostate cancer-specific mortality [22]. Pathological failure was defined as the presence of clinically significant PCa in biopsy. No salvage treatment was initiated in the absence of pathological failure.

### 2.5. Statistical Analysis

Continuous variables are reported as means ± standard deviations (SD) and medians with interquartile ranges (IQR) as appropriate, while categorical variables are presented as absolute numbers with percentages. All statistical analyses were performed using R software (version 4.0.2; R Foundation for Statistical Computing, Vienna, Austria).

## 3. Results

A total of 52 patients were enrolled. Data from one patient at a collaborating site were excluded from the analysis because of a major protocol deviation—his pre-treatment PSA level exceeded 15 ng/mL—leaving 51 patients in the per-protocol cohort. Mean follow-up time was 12.9 months (IQR 3–21)

The median age was 66.9 (IQR 60.5–73.5). The prostate volume was measured on MRI. ISUP Grade Group distribution was ISUP 1—20 Patients (39.2%), ISUP 2—25 Patients (49%), and ISUP 3—6 Patients (11.8%). Additional characteristics of the study cohort are shown in Table 1.

Treatment characteristics are summarized in Table 2.

Thirty-one patients (60.1%) underwent at least one post-HIFU biopsy; seven declined the protocol 12-month biopsy owing to persistently low, stable PSA.

Pathology was negative in 26/31 (83.9%). Cancer was found in five men (three ISUP 1, one ISUP 2, one ISUP 4). Among the three who ultimately required salvage therapy, all had non-palpable tumors (cT1c). Case 1 was graded ISUP 1 pre-HIFU but showed contralateral (out-of-field) ISUP 4 at the protocol control biopsy (12 months); an attempted salvage prostatectomy was aborted because of dense peri-rectal adhesions, and he received Radiation therapy (Rt). Case 2 had out-of-field ISUP 1 at the 12-month control biopsy; despite low-grade histology, PSA progression on follow-up led to Rt. Case 3 had in-field ISUP 2 on the first trigger-driven biopsy within the first year and proceeded to external-beam Rt (See Appendix A for the full case-by-case breakdown). Beyond 12 months, biopsies were performed only when predefined triggers suggested recurrence; among three such biopsies, two were negative and one showed ISUP 1 deemed clinically insignificant. Overall, three patients (5.9%) underwent definitive salvage treatment within 24 months; no metastases or prostate-cancer deaths occurred.

### 3.1. Safety

Procedure-related adverse events (AEs), defined as those considered related or possibly related to treatment, observed within the first 2 years are summarized in Table 3.

A total of 14 patients (27%) experienced at least one adverse event. The most common AE was transient acute urinary retention, occurring in 11 patients (21.6%), typically within 30 days following therapy. A single short hospital admission (2%) was recorded. Six patients (11.8%) required placement of a suprapubic catheter under local anesthesia. Three patients experienced urinary tract infections post-procedure; two of these cases involved epididymitis, which resolved with oral antibiotic therapy. By the 12-month follow-up, only one moderate AE remained unresolved—a recurrent urethral stricture. All other adverse events had resolved by the 3-month follow-up visit. No Clavien–Dindo Grade 4 adverse events were reported.

A single patient developed muscle-invasive bladder cancer 19 months after HIFU; definitive staging and treatment decisions were still pending at data cut-off. Because of the tumour’s anatomical location and the long latency, the event was considered unlikely to be attributable to the focal HIFU procedure.

Two patients died during follow-up from cardiovascular events—one from an ST-segment-elevation myocardial infarction (STEMI) and the other from a mitral valve rupture, occurring 7 month and 13 months after the last HIFU session, respectively. Neither death was judged related to the malignancy or to the HIFU treatment.

### 3.2. PSA Dynamics

Figure 1 illustrates the serum PSA trajectory for the entire study population (n = 51) over 24 months. Baseline mean PSA was 7.55 ng/mL (95% CI 6.57–8.53). A steep decline occurred within the first 3 months, reaching 2.31 ng/mL (95% CI 1.69–2.92); this represents an absolute reduction of 5.24 ng/mL and a relative drop of 69.3% from baseline.

Between 3- and 6-months PSA values stabilized at about 2.3 ng/mL (95% CI 1.56–3.16). A modest upward inflection was observed at 9 months (mean 2.87 ng/mL, 95% CI 1.91–3.84) and persisted through 12 months (2.93 ng/mL, 95% CI 1.43–4.42). Thereafter, PSA oscillated within a narrow band (2.4–2.9 ng/mL) with overlapping confidence intervals at 15, 18, 21 and 24 months, indicating biochemical steadiness rather than progressive rise. Notably, the upper 95% CI never exceeded 4.5 ng/mL during follow-up, suggesting durable PSA suppression for most patients.

Across ISUP 1 (n = 20), ISUP 2 (n = 25), and ISUP 3 (n = 6) patients, baseline mean PSA values were 7.3 ng/mL (95% CI 6.1–8.4), 7.2 ng/mL (5.8–8.5), and 9.4 ng/mL (6.6–12.2), respectively. All three groups showed a comparable early decline at 3 months (ISUP 1: 2.8 ng/mL; ISUP 2: 1.7 ng/mL; ISUP 3: 2.5 ng/mL) and maintained PSA levels within the 2–4 ng/mL range throughout 24 months. Confidence-interval widths overlapped at every time-point, and no sustained divergence of mean curves was observed (Figure 2).

### 3.3. Imaging Follow-Up

When the protocol was written, no validated mpMRI scoring system existed for the ablated prostate. Follow-up scans were therefore classified in binary fashion as “suspicious” or “not suspicious” for residual disease. Six scans were judged suspicious. Targeted biopsies confirmed cancer in two of those six cases—one out-of-field ISUP 4 lesion (described above) and one additional ISUP 1 focus—giving qualitative mpMRI a positive-predictive value of 33% in this cohort.

### 3.4. Functional Outcomes

When all 37 evaluable patients were pooled, mean IPSS rose transiently at the first post-treatment visit (median +1.8 points vs. baseline), indicating short-term irritative/obstructive symptoms. Thereafter the score declined steadily, falling below the pre-treatment mean by month 12 and remaining stable through month 24. Because baseline symptom burden strongly influences clinical interpretation, we stratified patients by their pre-treatment IPSS.

In the Good-baseline group IPSS remained within the minimal clinically important difference (±3 points) throughout follow-up, signaling preservation of already satisfactory urinary function.

Men with Moderate/Severe baseline symptoms experienced a clinically meaningful improvement: mean IPSS fell by 7.8 points (−59%) at 24 months, with the largest drop occurring between 6 and 12 months.

These divergent trajectories (illustrated in Figure 3) suggest that HIFU is function-sparing for patients who start with good LUTS profiles, while simultaneously offering symptom relief to those entering treatment with bothersome voiding complaints.

Mean baseline IIEF was 18.2 (95% CI 15.0–21.4). Scores dropped by roughly three points at 3 months (14.9; 11.0–18.8) but climbed to 17.1 at 6 months and 16.9 at 12 months. By 24 months the mean slightly surpassed baseline at 18.8 (14.2–23.3). Across all assessments the 95% confidence intervals overlapped the pre-treatment range, indicating only a transient, largely reversible decline in erectile function after therapy (Figure 4).

Only a single participant experienced de novo urinary incontinence, classified as grade 2 incontinence (loss of urine during physical activity or positional change). The episode followed a prolonged urinary-tract infection in a patient with poorly controlled diabetes and prior acute urinary retention.

### 3.5. Outcome of Quality of Life

Regarding health-related quality of life, 53% of patients completed the SF-36 questionnaire at scheduled follow-up visits. Adjusted mean scores at baseline and throughout follow-up were calculated using mixed-model analyses. Scores remained stable across all dimensions, with no statistically significant differences observed over time. Stability of scores suggests a sustained quality of life following the intervention. For clarity and improved readability, only data from pre-intervention, 12-month, and 24-month follow-up visits are illustrated in the radar chart (Figure 5A,B).

## 4. Discussion

This prospective study evaluated the medium-term oncological, functional and quality of life outcomes of HIFU in patients with localized prostate cancer. At 24 months, 94.1% of the patients avoided radical whole gland treatment. These real-life, multicenter, prospective results confirm the data from other international studies [23,24,25,26].

A key finding is the preservation of quality of life: patients treated with HIFU maintained stable health-related QoL scores throughout follow-up [27].

Urinary outcomes were particularly encouraging, which can be caused by the shrinkage of the prostate after the HIFU therapy [28,29].

Patients with good baseline function maintained stable, symptom-sparing outcomes, while those with bothersome LUTS experienced sustained relief, indicating that HIFU can both preserve and rehabilitate voiding function. The mechanism of symptom improvement in men with pre-existing LUTS remains to be clarified and may relate to treatment-induced changes in prostate tissue [29]. Recent updates in minimally invasive, thermally driven prostate procedures also underscore symptom benefits for LUTS in non-oncologic settings. In benign prostatic obstruction, transperineal laser ablation (TPLA) has shown consistent improvements in IPSS with low complication rates. While indications differ from focal HIFU for cancer, these data support the broader principle that targeted intraprostatic thermal ablation can relieve obstruction-related symptoms in appropriately selected patients [30].

The AE profile in our series closely paralleled that of earlier HIFU reports, both in frequency and in severity [7,23,31]. The two deaths observed during follow-up were attributed to pre-existing cardiovascular disease in older participants and were adjudicated as unrelated to either the cancer or the HIFU procedure.

Clinical practice continues to reveal two groups gravitating toward focal HIFU: younger patients focused on preserving erectile function and older, comorbidity-burdened individuals seeking to avoid the systemic stress of radical treatment [27,32,33]. The present findings indicate that HIFU meets the therapeutic priorities of both cohorts.

### Limitations

This study has several noteworthy constraints. First, enrollment and most follow-up visits took place during the COVID-19 pandemic; clinic-access restrictions meant that some assessments were captured at the nearest feasible visit rather than on the protocol date, introducing timing bias. Second, although the multicentre design enhances generalisability, it also led to minor inter-site differences in protocol adherence. Third, the trial was sized for feasibility, not for definitive oncological endpoints; with 51 evaluable patients and a median follow-up of 24 months, the study is under-powered to provide precise estimates of biochemical-recurrence-free, salvage-therapy-free, overall or cancer-specific survival.

Our diagnostic pathway relied on mpMRI with targeted plus systematic biopsies; however, mpMRI has an imperfect positive predictive value and sensitivity for clinically significant prostate cancer. Even with fusion-guided targeting, sampling error and MRI-occult foci can lead to under-grading or missed tumors. Such undetected or incompletely characterized lesions outside the ablation field can drive progression or salvage needs. Moreover, at the time of study design there was no consensus framework for interpreting post-HIFU mpMRI, and standard PI-RADS performs suboptimally in the ablation setting. A dedicated post-focal-therapy scoring system (PI-FAB) has since been proposed but our imaging was not prospectively graded using it [34].

Functional data carry additional caveats. Erectile-function outcomes relied solely on the patient-reported IIEF-5, and 32% of participants missed at least one IIEF-5 follow-up, widening confidence intervals and opening the door to responder bias. Overall questionnaire compliance fell from 100% at baseline to 63% at 24 months, leaving sizeable blocks of missing data. Ejaculatory function was not prospectively captured, so we cannot quantify ejaculatory preservation or changes after HIFU; this represents an important gap for future protocols.

Only 31 of 51 patients (61%) underwent the protocol-mandated 12-month biopsy, so residual disease may have been underestimated and the true positive-predictive value of mpMRI could not be fully assessed. Finally, in the absence of radical prostatectomy specimens, a definitive “true negative” cannot be established; our conclusions rest on biopsy, imaging, and biochemical endpoints rather than whole-mount histopathology.

The two-year horizon is too short to detect late oncological failures or delayed declines in urinary and sexual function. These limitations caution against over-interpreting the favourable early results and underscore the need for larger trials with longer, more complete follow-up.

## 5. Conclusions

In conclusion, this prospective multicentric 2-year follow-up study demonstrates that mpMRI-guided focal HIFU yields encouraging early oncologic outcomes with low genitourinary morbidity and preserved quality of life.

The authors would like to acknowledge all the teams that helped coordinate and execute this clinical study.

## Figures and Tables

**Figure 1 cancers-17-03429-f001:**
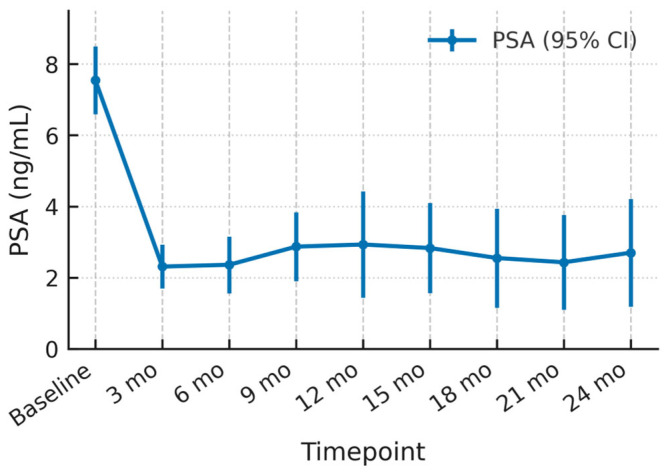
PSA dynamics following HIFU with 95% Confidence Intervals.

**Figure 2 cancers-17-03429-f002:**
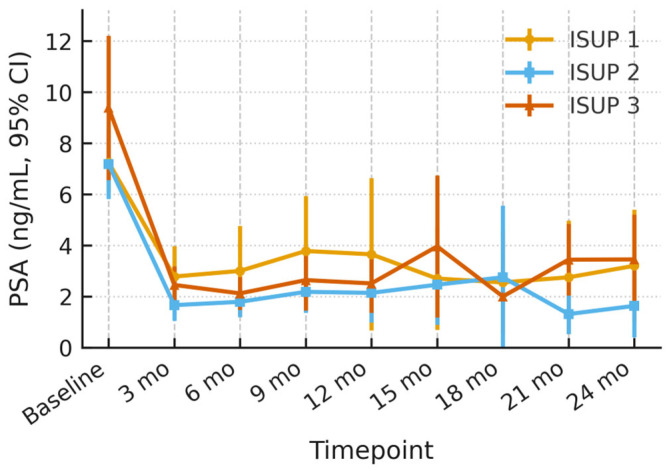
PSA Dynamics through the ISUP Subgroups after HIFU with 95% Confidence Interval.

**Figure 3 cancers-17-03429-f003:**
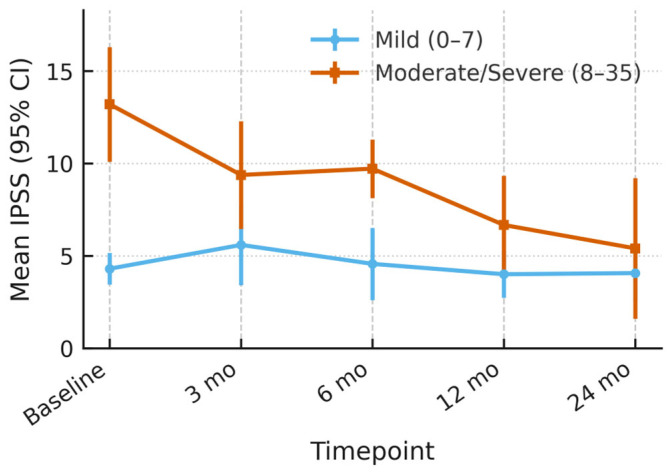
IPSS Trajectory after Therapy, stratified by baseline LUTS Severity.

**Figure 4 cancers-17-03429-f004:**
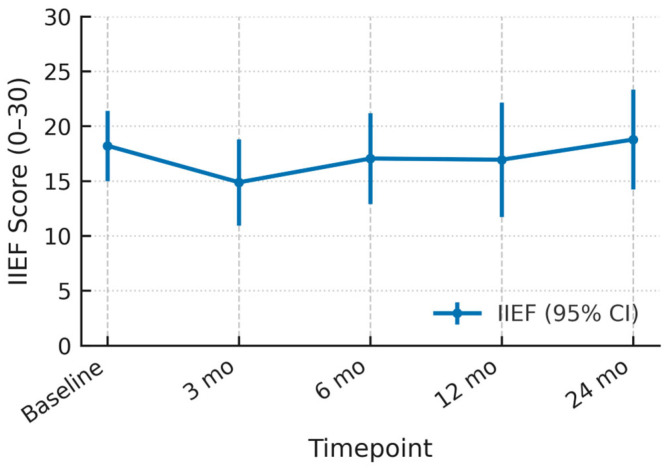
Mean IIEF Scores after HIFU with 95% Confidence Intervals.

**Figure 5 cancers-17-03429-f005:**
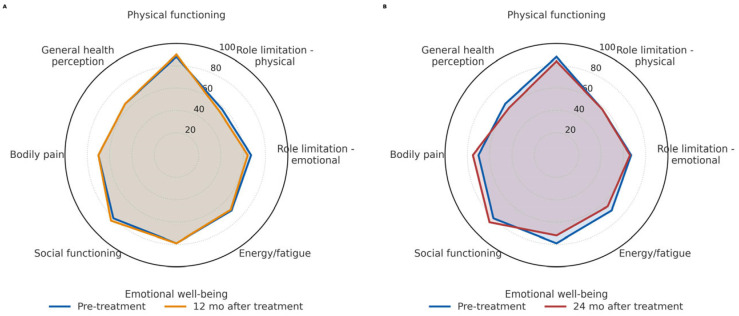
SF-36 domain scores displayed as radar charts: baseline vs. 12 months (**A**) and baseline vs. 24 months (**B**). Higher scores indicate better health-related quality of life.

**Table 1 cancers-17-03429-t001:** Baseline characteristics of the study cohort.

	Included Patients (n = 51)
Patient excluded	1
Age (years), median (IQR)	66.9 (60.5–73.5)
PSA (ng/mL), median (IQR)	7.55 (5.4–9.4)
Prostate volume, mL, median (IQR)	39.5 (29–50)
**ISUP**	
1	20 (38%)
2	25 (49%)
3	6 (12%)
**mpMRI lesion (PI-RADS score)**	
3	6 (12%)
4	28 (55%)
5	16 (31%)
**Clinical Stage**	
T1c	37 (73%)
T2	8 (16%)
**Medical history**	
5ARI	5 (10%)
Alpha-blocker	11 (22%)
Anticoagulation	18 (35%)
Prior prostate surgery (TUR/P)	1 (2%)
Charlson Comorbidity Index, mean (IQR)	4.2 (3–8)
Smoker (actively)	12 (24%)
Diabetes	9 (18%)
Positive family history of prostate cancer	5 (10%)

**Table 2 cancers-17-03429-t002:** Treatment parameters and peri-procedural details.

	All (n = 51)
Target volume, mL—median (IQR)	12.2 (9.9–13.0)
Procedure time, min—median (IQR)	84.6 (70.5–96.5)
Intraoperative CEUS application	18 (35%)
Foley catheter duration, days—median (IQR)	5 (3–5)

**Table 3 cancers-17-03429-t003:** Adverse Events.

	All (n = 51)
Any kind of AE	14 (27%)
Hospital admission ≤ 30 days post-HIFU	1 (2%)
**Transient acute urinary retention**	11 (22%)
With transurethral catheter placement	5 (10%)
With suprapubic catheter placement ^1^	6 (12%)
With post-HIFU TURP for deobstruction	2 (4%)
**Urinary Tract Infection**	3 (6%)
Epididymitis	2 (4%)
**Bladder Cancer**	1 (2%)
**Incontinence**	1 (2%)
**Death**	2 (4%)

^1^ in local anesthesia.

## Data Availability

Original data are available upon request from the corresponding authors.

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
