# Peer review of "Focal Therapy Using High-Intensity Focused Ultrasound for Low- and Intermediate-Risk Prostate Cancer: Results from a Prospective, Multicenter Feasibility Trial"

_cancers, 2025, doi:10.3390/cancers17213429_

Round 1

Reviewer 1 Report

Comments and Suggestions for Authors

Congrats to the authors on this well-written manuscript. The work is a prospective observational study testing feasibility in more than 50 men undergoing HIFU for prostate cancer in 3 Austrian centers. The authors evaluate multiple outcomes, including oncologic and functional outcomes, as well as safety.

Let me provide a few comments that may help improve the work: 

1) Title: include that it is a feasibility trial

2) Abstract: can omit the ethics protocol number, mention that it is a feasibility trial with (preliminary) assessments of patient response to HIFU, can remove this: “a 9 mm safety margin was applied; procedures were performed under general anaesthesia”, would just mention that 51 men were analysed in the per-protocol cohort (omitting that 52 patients were enrolled; one violated eligibility criteria); would add the mean PSA at 24 months (+/- SD).

3) Introduction: remove the dot after PCa before reference [6]; in the last paragraph of the introduction, formulate the aim of the study more carefully, as the study was not powered to assess oncologic outcomes.

4) Materials and Methods: please add at least the months of study conduction (mm.2021 – mm.2024), please adjust the wording of the eligibility criteria: per D’Amico, intermediate-risk would include patients with PSA up to 20 ng/mL; would suggest using Gleason grade group instead ISUP (in the whole body of the work); change “treatment pass“ to treatment; mention that it is a feasibility trial with (preliminary) assessments of patient response to HIFU – please also write out FFS when using it the first time in the manuscript; would mention imaging modalities for assessment of metastases, and change “pathological fail“ to pathological failure. Please also add that you were calculating means and SD when appropriate.

5) Results: Please add SD to the mean follow-up time. Revise the (whole manuscript and tables) for spelling, i.e., Prostate Volume in the text should be prostate volume. Also, no need to mention prostate size or ISUPs in text, as it is readily available in Table 1. Would make the denominator n=51 in Table 1, as this was the number of included patients. Please consider inserting a header within Table 1 for 5ARI, alpha-blocker, and anticoagulation. Indicate if the 12 smokers were actively smoking at the time of enrollment. Also, please add contralateral mpMRI lesion (PI-RADS score) to Table 1. Can write T1c instead of “cT1c” as there is the header Clinical Stage. Please clarify in Table 2 if this is the median or mean targeted volume, and add that you list the mean duration of Foley catheter in days. Why were only 60.1 % of the treated cohort undergoing at least one post-HIFU biopsy? Was the first post-HIFU biopsy not mandated by the protocol? Please spell out "Rt" when you use it for the first time in the manuscript. Insert a Title for Table 3, and please clarify: was SPT placement and TURP required for retention? Would recommend to rather list conditions here: urinary retention, the reason for hospital admission, UTI, incontinence, and death. For all figures, please elaborate better on the figure and table legends and explain what is seen to the readers. In terms of collaboration with the radiology department, it is unclear who performed the radiology reads – you may consider clarifying with authorship or acknowledgment. Please include the size of the prostate after the HIFU therapy, and if you saw an association of prostate size and LUTS/IPSS. Figure 3: would do the cut off at 8 with IPSS 0-7 representing mild LUTS.

6) Discussion: shorten this section: “Good baseline function (IPSS ≤ 8)—scores remained within the minimal clinically important difference throughout follow-up, indicating true function-sparing. Moderate/severe baseline LUTS (IPSS > 8)—mean IPSS fell from 13.1 to 5.8 (-45 %) by 24 months, suggesting that prostate shrinkage after focal therapy can translate into durable symptom relief”.

Please also include limitations with PI-RADS interpretation post-HIFU, and include potential literature that may be available. Also, with only about 60% undergoing post-HIFU biopsy, results need to be interpreted carefully. It should be noted that in the absence of radical prostatectomy, a definitive true negative cannot be established (lack of histopathological confirmation).

Conclusions: Please insert a comma after: "In conclusion"

I would like to thank the authors once again for their great effort in conducting this study, which I can imagine was particularly challenging in ensuring adherence during the COVID-19 pandemic. This work meaningfully enriches the urologic literature and presents promising results in patients with non-aggressive prostate cancer, warranting further investigation in larger populations.

Reviewer 2 Report

Comments and Suggestions for Authors

The authors aimed to evaluate oncological control, functional outcomes and safety of MRI-guided focal HIFU in patients with low- or intermediate-risk PCa.

They relied on phase II trial involving 51 patients. The methodology is robust.

Briefly, they reported at 24 months, failure-free survival (FFS) of 94.1 %: 3/51 patients (5.9 %) required salvage radiotherapy. Transient acute urinary retention occurred in 11/51 (21.6 %); no Clavien–Dindo grade ≥4 events were reported. IPSS returned to or improved beyond baseline, erectile function largely recovered by 6-12 months, and only one new case of grade 2 incontinence was observed.

The methodology is robust. However, several comments should be raised:

  • How many CD complications greater than 3 were reported? Why did the authors choose a cut-off at 4?
  • Also ISUP 3 patients were included. Why? This is not the current clinical practise.
  • Which were the characteristics of patients in which HIFU failed? ISUP, PSA, cT? number of cores?
  • Were the patients previously treated with other intervention (TURP, Holep, Rezum)? It should be ackowledge. Moreover, the role of incidental prostate cancer and focaltherapy should be discussed.
  • Which was the rates of ejaculatory preservation? It should be acknowledged. The current paper should be discussed (PMID 40548982)

Round 2

Reviewer 1 Report

Comments and Suggestions for Authors

Thanks to the authors for revising the manuscript!

Author Response

Thank you again for your comments and for the constructive review process!

Reviewer 2 Report

Comments and Suggestions for Authors

Recent updates on the topic should be described as well (PMID 40548982)

Author Response

Thank you for the suggestion to include recent updates! Very interesting article about this novel therapy. 

We have added a brief paragraph in the Discussion summarizing contemporary evidence on transperineal laser ablation for LUTS/BPH—highlighting consistent IPSS improvements with low complication rates, to contextualize our HIFU findings within the broader landscape of prostate-directed thermotherapies. 
(Page 9)